# Interests and Curiosities about Sexuality of Children and Adolescents from Northern Portugal

**Zélia Caçador Anastácio** [1,*], **Regina Ferreira Alves** [1], **Celeste Antão** [2], **María Dolores Gil-Llario** [3] **and Rafael Ballester-Arnal** [4]

1   Child Studies Centre (CIEC), Institute of Education, University of Minho, 4710-057 Braga, Portugal; rgnalves@gmail.com
2   Health Sciences Research Unit: Nursing (UICISA: E), Nursing School of Coimbra (ESEnfC), Polytechnic Institute of Bragança, 5300-253 Bragança, Portugal; celeste@ipb.pt
3   Department of Developmental and Educational Psychology, Faculty of Psychology, University of Valencia, 46010 Valencia, Spain; dolores.gil@uv.es
4   Department of Basic and Clinical Psychology and Psychobiology, Faculty of Health Sciences, Jaume I University, 12007 Castellón de la Plana, Spain; rballest@uji.es
*   Correspondence: zeliaf@ie.uminho.pt

**Abstract:** Sexuality education is a part of the health education process in schools. However, many programs of sexuality education do not consider the needs of children and adolescents. This study is based on an analysis of the interests of children and adolescents about sexuality. The sample covered 32 classes from primary school to higher education in the northern region of Portugal. The methodology was mixed, collecting data through open questions. It used an A5 white page, containing only options to indicate age, school grade, and sex. In the white space, students wrote their questions/doubts. A database was built in the program SPSS and categories were established a priori following the key concepts for sexuality education defined by UNESCO. A pattern of issues and interests was found depending on the age group and sex, noting that the children's interests are related to conception, birth, and the well-being of the fetus/newborn. Adolescents' interests are focused on contraception, sexually transmitted infections, and relationships. This study can help teachers to promote sexuality education which is appropriate to the developmental stage of their students, motivating them to better learn and leading to a reduction in risky sexual behaviors and conscious decision making for healthy sexuality.

**Keywords:** students' sexual curiosity; health promotion; sexuality education





## 1. Introduction

In the European context, for sexuality education in schools, guidelines can be found from UNESCO [1], WHO [2], and the network Schools for Health in Europe [3], encompassing topics such as sexual development, sexual and reproductive health and rights, intimacy, consent, body image, and gender roles. At a national level, governmental laws and norms, as well as guidelines from the Portuguese Ministry of Education [4], can also be used to help in the implementation of sexuality education in schools. These guidelines present the topics to be approached and the learning objectives by age group [1–3] or school grade [4]. Despite the guidelines, many sexuality education programs implemented in the school context do not consider the needs, interests, curiosities, and doubts of the students. Additionally, if we search the literature about how to approach sexuality issues in children [5–7], namely strategies to respond to questions about babies and genital names, there is not a full list of the types of questions and doubts of children and adolescents regarding their developmental stages. Some research on the type of sexuality-related questions according to developmental stage can be found in work by Ariza et al. [8] and Barragán [9] from

Spain. Thus, this reveals the need for research to plan sexuality education programs in schools that are appropriate for the student's interests and answer their questions.

On the other hand, in recent years, the scientific aspects of human sexuality have been underestimated, to the detriment of the psychological and social dimensions [10], due to the harsh criticisms that have been made of the approach that reduces the biological educational aspect, associated with the preventive medical health education model. Another trend observed is that the topics of sexuality education have been changing over the decades according to educational and public health priorities, although many themes remained unchanged. During the 1960s–1970s, there was a concern with the prevention of unwanted pregnancies, in the late 1980s with HIV prevention, in the 1990s with sexual abuse, and from the 2000s with the prevention of sexism, homophobia, and bullying. Currently, sexuality education is very much concerned with addressing gender stereotypes and inequalities [11]. Accordingly, sexual health education should be recognized as a political, sociocultural, and educational process.

Different concepts appear in the literature concerned with educational actions centered on sexuality, namely sex education, sexual education, sexuality education, and, more recently, comprehensive sexuality education. We opted for the term sexuality education since it is the most used by UNESCO [1] and the WHO [2].

Sexuality education is a formal means through which adolescents are introduced to the biopsychosocial aspects of sexuality and sexual health [12]. Despite the prevalence and diversity of adolescent's sexual experiences, different views on the nature of sexuality education and what is considered appropriate or acceptable in schools has led to inconsistency in the implementation of sexuality education programs in schools.

Sexuality education is a topic in health education in schools, and health education projects usually start with an analysis of the needs and interests of the participants in this context [13]. In this way, this study is the first study at an international level that compares the interests, doubts, curiosities, and questions of children and adolescents at school, with respect to their phase of development and the international guidelines for the implementation of a comprehensive sexuality education in schools [1]. Another study with a similar research design based on anonymous questions about sexuality in a project on Health Education for Youth was carried out in the USA [14].

This research aimed to conduct a survey of students' interests, curiosities, and questions about human sexuality in each school grade and to establish a pattern of questions according to their developmental stage, considering the different domains of sexuality education based on the key concepts defined internationally. In this sense, the main research question formulated for the purpose of this study was: In which areas of sexuality education do students have more interest and curiosity? As sub-questions, that can be also transformed into specific objectives, we want to know (1) what is the extent of interest in issues related to biology and/or natural science education or psychology and social sciences; (2) if girls and boys have equal or different curiosities; and (3) if the interests in the sexuality domain vary according to school level. The following null hypotheses are formulated: $H_{01}$—there are no differences between boys and girls in sexuality questions and $H_{02}$—the interests in sexuality are different according to the developmental stage and school grade.

## 2. Materials and Methods

This was a transversal and descriptive study. Data were collected near each participant at a single point in time. The methodology was mixed, since data were collected by closed and open questions. Interpretation of the content was performed for data analysis. This type of methodology was chosen for this research because the data collected were questions written openly by the individuals, which were then targeted by content analysis to perform categorization. We used the exclusive categorization of the content units, and after that, we performed a quantification of the obtained categories, as recommended in [15,16], in order to verify the predominant occurrences.

*2.1. Procedures for Data Collection and Categorizations*

To identify the needs and questions of children and adolescents about sexuality in a school context, we opted for the collection of students' questions anonymously. This method has been widely used in sexuality education strategies and is called the "question box". Stevens and colleagues [14] reported that the use of a box of anonymous questions written by young people is a safe alternative strategy for participants to place questions concerning human sexuality, as many participants in sexuality education lessons may feel uncomfortable asking questions orally.

Data collection was performed using an A5 white sheet of paper (containing only options to indicate the individual variables of age, school grade, and sex), where students wrote their questions or doubts in the context of the classroom. Students were asked to fill in the individual variables and to write the questions they had concerning sexuality. Then, a database was built in SPSS (version 28.0), containing a transcription of all the questions written by the children and adolescents. To identify the relation between the categories of questions and the individual factors, sexes, and school grades, we applied the Chi-Square test.

For data analysis, since we intended to classify the questions according to the areas defined in the guidelines from UNESCO [1], we established categories a priori, performing a pre-ordinate categorization [16]. The categories defined were the eight key concepts for sexuality education defined by UNESCO [1] and the subcategories were the topics to be approached in each key concept, as follows:

1.  Relationships: 1.1. Families; 1.2. Friendship, Love, and Romantic Relationships; 1.3. Tolerance, Inclusion, and Respect; 1.4. Long-term Commitments and Parenting.
2.  Values, Rights, Culture, and Sexuality: 2.1. Values and Sexuality; 2.2. Human Rights and Sexuality; 2.3. Culture, Society, and Sexuality.
3.  Understanding Gender: 3.1. The Social Construction of Gender and Gender Norms; 3.2. Gender Equality, Stereotypes, and Bias; 3.3. Gender-based Violence.
4.  Violence and Staying Safe: 4.1. Violence; 4.2. Consent, Privacy, and Bodily Integrity; 4.3. Safe use of Information and Communication Technologies (ICTs).
5.  Skills for Health and Well-being: 5.1. Norms and Peer Influence on Sexual Behavior; 5.2. Decision making; 5.3. Communication, Refusal, and Negotiation Skills; 5.4. Media Literacy and Sexuality; 5.5. Finding Help and Support.
6.  The Human Body and Development: 6.1. Sexual and Reproductive Anatomy and Physiology; 6.2. Reproduction; 6.3. Puberty; 6.4. Body Image.
7.  Sexuality and Sexual Behavior: 7.1. Sex, Sexuality, and the Sexual Life Cycle; 7.2. Sexual Behavior and Sexual Response.
8.  Sexual and Reproductive Health: 8.1. Pregnancy and Pregnancy Prevention; 8.2. HIV and AIDS Stigma, Care, Treatment, and Support; 8.3. Understanding, Recognizing, and Reducing the Risk of STIs, including HIV.

Given the data obtained, the need to create two additional categories emerged, labeled "out of context" and "no reply", which led to 10 categories. The "no reply" category was also considered for analysis as it can be an important indicator [15,16] and because we observed in sexuality education sessions that in general girls ask more questions than boys. In addition, some questions were not appropriate to be included in the previously defined subcategories. In these cases, we created new subcategories a posteriori, which can be identified in the tables of results in the next section. To increase the validity of the categorization, it was performed by three researchers separately, and afterwards, the researchers met to compare the categories each one obtained. No great differences were observed between the three categorizations. When some questions were categorized differently, researchers decided on the category in this meeting based on the description of the UNESCO key concepts [1].

*2.2. Sample*

The sample covered a total of 741 students, 353 (47.7%) females and 387 (52.3%) males, with an average age of 13.61 years old. Students were from 32 classes from primary school to higher education, in the northern region of Portugal, and data collection was planned in 2018 and occurred between 2019 and 2020. According to the school level, 134 participants were in primary school (or Portuguese first cycle of basic education—CBE, 1st to 4th school grade), 139 were in the 2nd CBE (5th to 6th grade), 160 in the 3rd CBE (7th to 9th grade), 148 in secondary school (10th to 12th grade), and 160 in higher education. We obtained a total of 1453 questions, of which 277 were not related to sexuality issues. Thus, we received 1176 sexuality-related questions. A total of 169 students filled in the characterization data but did not write any questions. It is also worth pointing out that in the vast majority of cases, each student drafted more than one question.

*2.3. Ethical Procedures*

This study is part of a whole action research project, more precisely the needs identification or diagnosis step. All the ethical procedures were respected according to the code of ethical conduct of the University of Minho for scientific research, having obtained approval from the Ethical Committee for Research in Human and Social Sciences (CEICSH 007/2020). Data collection was totally anonymous, participants were informed about the aim of the study, participation was voluntary without coercion or rewards for participation, and participants could withdraw at any moment. The Helsinki principles for research with humans were adhered to, as well as the Portuguese Law for research and data protection. Before the project activities, teachers and school principals gave consent and informed the parents or guardians of children and adolescents less than 18 years old. In addition, it is important to note that sexuality education is compulsory in Portugal. The best interests of children were always considered.

**3. Results**

*3.1. Categories of Question, by Sexes and by Developmental Stage/School Grade*

We focused our content analysis on the categories classified according to the eight key concepts of UNESCO [1] for sexuality education. The results will follow our research question. Firstly, to identify the areas of sexuality education in which students have more interests and curiosities, a data analysis of the 1176 sexuality-related questions showed a great interest related to the category "The Human Body and Development" (39.0%), followed by the categories "Sexual and Reproductive Health" (26.4%) and "Sexuality and Sexual Behavior" (26.4%). Lower percentages were observed for "Understanding gender" (4%), "Skills for Health and Well-being" (1.9%), and Relationships" (1.4%). A very reduced interest was observed in the other two categories: "Values, Rights, Culture, and Sexuality" (0.7%) and "Violence and Staying Safe" (0.1%). It can be observed that the three most frequent categories "The Human Body and Development", "Sexuality and Sexual Behavior", and "Sexual and Reproductive Health" account for 91.8% of the sexuality-related questions.

Table 1 shows the distribution of all categories, considering all the answers, including those out of context and those with "no reply", by sex. A detailed analysis reveals a greater female interest related to the categories "The Human Body and Development" and "Sexual and Reproductive Health" compared to males. Boys tend to show more curiosity towards issues concerned with "Sexuality and Sexual Behavior". In addition, boys were more likely to refuse to ask questions, i.e., the category "no reply". Differences between sexes were statistically significant ($\chi^2$ (9) = 75.835, $p < 0.001$) and the effect size (V = 0.216) is near the medium [16].

**Table 1.** Frequencies and percentages for all categories by sexes.

| Categories | Female | | Male | |
|---|---|---|---|---|
| | F | % | F | % |
| 1. Relationships | 8 | 1.0% | 8 | 1.0% |
| 2. Values, Rights, Culture, and Sexuality | 4 | 0.5% | 4 | 0.5% |
| 3. Understanding Gender | 22 | 2.8% | 25 | 3.0% |
| 4. Violence and Staying Safe | 1 | 0.1% | 0 | 0.0% |
| 5. Skills for Health and Well-being | 8 | 1.0% | 15 | 1.8% |
| 6. The Human Body and Development | 275 | 34.4% | 183 | 22.3% |
| 7. Sexuality and Sexual Behavior | 114 | 14.3% | 197 | 24.0% |
| 8. Sexual and Reproductive Health | 179 | 22.4% | 132 | 16.0% |
| Out of Context | 136 | 17.0% | 141 | 17.2% |
| No Reply | 52 | 6.5% | 117 | 14.2% |
| Total | 799 | 100.0% | 822 | 100.0% |

F = Frequency; % = Percentage.

We also sought to find out the relationship between the categories of questions and the school level. We observed that the children of 1st and 2nd CBE were those who asked more questions out of context, as well as those who wrote more questions related to "The Human Body and Development". Primary school children (1st CBE) were less interested in questions related to the category "Sexual and Reproductive Health". This category was highlighted for the 2nd CBE and the 3rd CBE (age range 10–15), consisting of children/pre-adolescents and adolescents, and registered more questions from secondary school adolescents. Questions in the category "Sexuality and Sexual Behavior" were mainly asked by students in the 3rd CBE and secondary school, although they were also frequent for the 2nd CBE. Adolescents attending secondary school asked more questions about "Sexual and Reproductive Health" and "Sexuality and Sexual Behavior". Students attending higher education (HE) registered more questions for the "The Human Body and Development" category. Students from secondary school and higher education registered more "No replies". This distribution of frequencies can be observed in Table 2.

**Table 2.** Frequencies for categories by school grade.

| Categories | 1st CBE | 2nd CBE | 3rd CBE | SecSc | HE | Total |
|---|---|---|---|---|---|---|
| 1. Relationships | 1 | 9 | 2 | 4 | 0 | 16 |
| 2. Values, Rights, Culture, and Sexuality | 0 | 2 | 3 | 1 | 2 | 8 |
| 3. Understanding Gender | 5 | 22 | 14 | 4 | 2 | 47 |
| 4. Violence and Staying Safe | 0 | 1 | 0 | 0 | 0 | 1 |
| 5. Skills for Health and Well-being | 5 | 8 | 6 | 2 | 2 | 23 |
| 6. The Human Body and Development | 146 | 162 | 57 | 24 | 70 | 459 |
| 7. Sexuality and Sexual Behavior | 6 | 52 | 186 | 54 | 13 | 311 |
| 8. Sexual and Reproductive Health | 3 | 115 | 86 | 57 | 50 | 311 |
| Out of Context | 68 | 164 | 32 | 6 | 7 | 277 |
| No Reply | 1 | 1 | 23 | 55 | 89 | 169 |
| Total | 235 | 536 | 409 | 207 | 235 | 1622 |

CBE = Cycle of Basic Education; SecSc = Secondary School; HE = Higher Education.

Searching for dependency relationships between the categories of questions and the school grade, the Chi-square test indicated a significant relationship ($\chi^2$ (36) = 908.675, $p < 0.001$) and effect size (V = 0.374) above the medium [16].

Questions considered as out of context were more frequent in 2nd CBE and, in general, were related to animals (analogies about birth and reproductions; science; drug dependency, alcohol, and tobacco; several diseases, but more about cardiovascular diseases and cancers (unrelated to reproductive and sexual health)).

### 3.2. Subcategories of Sexuality Questions

Since the focus of this research is on sexuality-related questions, the subcategory analysis follows without the questions categorized as "out of context" and the "no reply". The table of topics from UNESCO [1] for the categorization a priori did not contain enough categories to classify all the questions obtained. In this sense, some additional subcategories were included, as Table 3 shows in bold. The organization of the questions in the subcategories showed greater interest in "Sexual and Reproductive Anatomy and Physiology" and "Reproduction" (in the category Human Body and Development), "Sex, Sexuality, and the Sexual Life Cycle", and "Sexual Behavior and Sexual Response" (in the category Sexuality and Sexual Behavior), as well as in the new subcategory "Contraceptive methods", created for the category Sexual and Reproductive Health.

**Table 3.** Frequencies of categories and subcategories by sexes.

| Categories | Subcategories | Female | Male | Total |
|---|---|---|---|---|
| 1. Relationships | | 8 | 8 | 16 |
| | 1.1. Families | 0 | 0 | 0 |
| | 1.2. Friendship, Love, and Romantic Relationships | 5 | 8 | 13 |
| | 1.3. Tolerance, Inclusion, and Respect | 0 | 0 | 0 |
| | 1.4. Long-term Commitments and Parenting | 3 | 0 | 3 |
| 2. Values, Rights, Culture and Sexuality | | 4 | 4 | 8 |
| | 2.1. Values and Sexuality | 4 | 0 | 4 |
| | 2.2. Human Rights and Sexuality | 0 | 3 | 3 |
| | 2.3. Culture, Society, and Sexuality | 0 | 1 | 1 |
| 3. Understanding Gender | | 22 | 25 | 47 |
| | 3.1. The Social Construction of Gender and Gender Norms | 3 | 1 | 4 |
| | 3.2. Gender Equality, Stereotypes, and Bias | 14 | 12 | 26 |
| | 3.3. Gender-based Violence | 3 | 4 | 7 |
| | **3.4. Sexual Orientation** | 2 | 8 | 10 |
| 4. Violence and Staying Safe | | 1 | 0 | 1 |
| | 4.1. Violence | 1 | 0 | 1 |
| | 4.2. Consent, Privacy, and Bodily Integrity | 0 | 0 | 0 |
| | 4.3. Safe use of Information and Communication Technologies (ICTs) | 0 | 0 | 0 |
| 5. Skills for Health and Well-being | | 8 | 15 | 23 |
| | 5.1. Norms and Peer Influence on Sexual Behavior | 1 | 1 | 2 |
| | 5.2. Decision-making | 0 | 0 | 0 |
| | 5.3. Communication, Refusal, and Negotiation Skills | 0 | 0 | 0 |
| | 5.4. Media Literacy and Sexuality | 1 | 2 | 3 |
| | 5.5. Finding Help and Support | 6 | 12 | 18 |
| 6. The Human Body and Development | | 275 | 183 | 459 |
| | 6.1. Sexual and Reproductive Anatomy and Physiology | 167 | 83 | 250 |
| | 6.2. Reproduction | 89 | 79 | 168 |
| | 6.3. Puberty | 14 | 13 | 27 |
| | 6.4. Body Image | 5 | 8 | 13 |
| 7. Sexuality and Sexual Behavior | | 114 | 197 | 311 |
| | 7.1. Sex, Sexuality, and the Sexual Life Cycle | 50 | 105 | 155 |
| | 7.2. Sexual Behavior and Sexual Response | 64 | 92 | 156 |

**Table 3.** *Cont.*

| Categories | Subcategories | Female | Male | Total |
|---|---|---|---|---|
| 8. Sexual and Reproductive Health | | 179 | 132 | 311 |
| | 8.1. Pregnancy and Pregnancy Prevention | 41 | 16 | 57 |
| | 8.2. HIV and AIDS Stigma, Care, Treatment, and Support | 29 | 19 | 48 |
| | 8.3. Understanding, Recognizing, and Reducing the Risk of STIs, including HIV | 15 | 14 | 29 |
| | **8.4. Abortion** | 21 | 15 | 36 |
| | **8.5. Contraceptive methods** | 60 | 61 | 121 |
| | **8.6. Cancer** | 13 | 7 | 20 |
| Total | | 611 | 564 | 1176 |

Bold format highlights the new subcategories created.

Presenting the categorization by each category and respective subcategories, we now present the frequency of questions accompanied by some examples to elucidate them. The sentences are followed by an indication of sex (F = Female or M = Male), age (numeric), and school level (1st CBE, 2nd CBE, 3rd CBE, secondary school, or HE). For the category "Relationships", the subcategory "Friendship, love, and romantic relationships" registered the majority of occurrences (Table 4).

**Table 4.** Frequencies and examples of questions for the subcategory of Relationships.

| Category | Subcategories | F |
|---|---|---|
| 1. Relationships | | 16 |
| | 1.2 Friendship, love, and romantic relationships | |
| | Ex: Why would a girl or a boy fall in love so young? (F, 11, 2nd CBE) | 13 |
| | Ex: What gift can be given to a girl? What kind of gifts? (M, 15, 2nd CBE) | |
| | 1.4 Long-term commitment, marriage, and parenting | |
| | Ex: Why do women marry men if there is betrayal between them? (F, 11, 2nd CBE) | 3 |
| | Ex: Why would somebody have children and then abandon or kill them? (F, 11, 2nd CBE) | |

For the category "Values, Rights, Culture, and Sexuality", the subcategory "Values and Sexuality" registered more occurrences (Table 5). For the subcategory "Human Rights and Sexuality", the three questions were related to abortion and concerned with being in favor or against it, which is about rights, not about its consequences for sexual and reproductive health.

**Table 5.** Frequencies and examples of questions for the subcategory of Values, Rights, Culture, and Sexuality.

| Category | Subcategories | F |
|---|---|---|
| 2. Values, Rights, Culture, and Sexuality | | 8 |
| | 2.1. Values and sexuality | |
| | Ex: Why can't priests get married? After all they are men, and I guess any man should have "that desire"? (F, 11, 2nd CBE) | 4 |
| | Ex: Why is there a social stigma that represses underage single mothers? (F, 18, HE) | |
| | 2.2. Human rights and sexuality | |
| | Ex: I am in favor of abortion, because each person knows about his or her own life. (M, 14, 3rd CBE) | 3 |
| | 2.3. Culture, society, and sexuality | |
| | Why are some people traumatized by being sexually harassed? (M, 15, secondary school) | 1 |

For the category "Understanding Gender", the subcategory "Gender Equality, Stereotypes, and Bias" was the most frequent (Table 6). In addition, an important frequency of questions related to the Sexual Orientation subcategory emerged, especially from students in 2nd CBE.

**Table 6.** Frequencies and examples of questions for the subcategory of Understanding Gender.

| Category | Subcategories | F |
|---|---|---|
| 3. Understanding Gender | | 47 |
| | 3.1. The Social Construction of Gender and Gender Norms | 4 |
| | Ex: Can a boy in the middle of his life look like a girl? (F, 9, 1st CBE) | |
| | Why are some children girls inside and boys outside? (M, 12, 2nd CBE) | |
| | 3.2. Gender Equality, Stereotypes, and Bias | 26 |
| | Ex: Why do boys like sex more than girls? (F, 12, 3rd CBE) | |
| | Ex: Why do women not like porn videos? (M, 18, HE) | |
| | 3.3. Gender-based Violence | 7 |
| | Ex: Why do ugly girls not pay a fine because they are so ugly and boring and horrible? (M, 12, 2nd CBE) | |
| | Ex: Why do boys sometimes force girls into less-than-decent things? (F, 13, 3rd CBE) | |
| | **3.4. Sexual Orientation** | 10 |
| | Ex: Why are men gay? (M, 12, 2nd CBE) | |
| | Ex: How could two women make love? (F, 12, 2nd CBE) | |

Bold format highlights the new subcategories created.

The category "Violence and Staying Safe" included only one question (Ex: Why there are so many women being raped in the world? (F, 11, 2nd CBE)) in the subcategory "Violence".

Questions in the category "Skills for Health and Well-being" were more associated with "Finding Help and Support" (Table 7), namely about sexual education at school, a question that was repeated several times. In addition, it was more frequent among boys, as shown above in Table 3.

**Table 7.** Frequencies and examples of questions for the subcategory of Skills for Health and Well-being.

| Category | Subcategories | F |
|---|---|---|
| 5. Skills for Health and Well-being | | 23 |
| | 5.1. Norms and peer influence on sexual behaviour | 2 |
| | Ex: Is having a boyfriend at the age of fifteen good or bad? Is it a good or bad influence? (F, 12, 2nd CBE) | |
| | 5.4. Media literacy and sexuality | 3 |
| | Ex: Why are we curious to see pornographic films? (M, 12, 2nd CBE) | |
| | 5.5. Finding Help and Support | 18 |
| | Ex: Why is there no room at school to clarify our doubts about life? (M, 11, 2nd CBE) | |
| | Ex: Why are there no sexual education classes? (F, 11, 2nd CBE) | |

In the category "The Human Body and Development", the most frequent subcategory was "Sexual and Reproductive Anatomy and Physiology", followed by the subcategory "Reproduction". These topics are very interesting for primary school children. "Puberty" is also an interesting issue for students in their developmental stage (Table 8). These three subcategories had more questions from girls than boys, as presented in Table 3 above. Boys asked more questions than girls only for the subcategory "Body image".

**Table 8.** Frequencies and examples of questions for the subcategory of The Human Body and Development.

| Category | Subcategories | F |
|---|---|---|
| 6. The Human Body and Development | | 459 |
| | 6.1. Sexual and Reproductive Anatomy and Physiology | |
| | Ex: How do sperm join with the cell? (F, 9, 1st CBE) | 250 |
| | Ex: How do men's testicles produce sperm? (F, 11, 2nd CBE) | |
| | 6.2. Reproduction | |
| | Ex: How does a baby appear in the belly? (M, 7, 1st CBE) | 168 |
| | Ex: How does a baby eat inside the belly of the mother? (F, 8, 1st CBE) | |
| | 6.3. Puberty | |
| | Ex: I would like to know more about menstruation. (F, 10, 2nd CBE) | 27 |
| | Ex: Why does hair appear in the armpits and pubic region in adolescence? (F, 12, 2nd CBE) | |
| | 6.4. Body Image | |
| | Ex: If we are ugly, when we grow up will we be more beautiful? (M, 11, 2nd CBE) | 13 |

The category "Sexuality and Sexual Behavior" was one of the second most frequent. The two subcategories were considered equally important by students over 10 years old (Table 9).

**Table 9.** Frequencies and examples of questions for the subcategory of Sexuality and Sexual Behavior.

| Category | Subcategories | F |
|---|---|---|
| 7. Sexuality and Sexual Behavior | | 311 |
| | 7.1. Sex, Sexuality, and the Sexual Life Cycle | |
| | Ex: How old can we start having sex? (M, 13, 2nd CBE) | 155 |
| | Ex: Why must people aged 60 use Viagra? (M, 13, 2nd CBE) | |
| | Ex: Why can't the elderly have sex? (F, 12, 2nd CBE) | |
| | 7.2. Sexual Behaviour and Sexual Response | |
| | Ex: How can we control sexual desire? (M, 12, 3rd CBE) | 156 |
| | Ex: What is an orgasm? (M, 12, 3rd CBE) | |
| | Ex: What is premature ejaculation? How can we cure it? (F, 12, 3rd CBE) | |
| | Ex: How do you lose your virginity? (F, 15, secondary school) | |

In the category "Sexual and Reproductive Health", the emerged subcategory "Contraceptive methods" was the most frequent. We received many questions, both from girls and boys, such the examples shown (Table 10). The topic of the risk of sexually transmissible infections (STIs) is also present in a considerable number of questions. The other two subcategories that emerged were abortion and cancer.

**Table 10.** Frequencies and examples of questions for the subcategory of Sexual and Reproductive Health.

| Category | Subcategories | F |
|---|---|---|
| 8. Sexual and Reproductive Health | | 311 |
| | 8.1. Pregnancy and Pregnancy Prevention | |
| | Ex: Is it dangerous to take the emergency pill? Does it cause any diseases? (F, 13, 3rd CBE) | 57 |
| | Ex: When a person has sexual intercourse without using a condom, does the woman necessarily have a baby, or is this not always the case? (M, 11, 2nd) | |

**Table 10.** *Cont.*

| Category | Subcategories | F |
|---|---|---|
| | 8.2. HIV and AIDS Stigma, Care, Treatment, and Support | 48 |
| | Ex: How AIDS is detected? What are the symptoms? (F, 14, 3rd CBE) | |
| | 8.3. Understanding, Recognizing and Reducing the Risk of STIs, including HIV | 29 |
| | Ex: What are the most serious sexually transmitted diseases? (M, 18, HE) | |
| | Ex: When the partner has herpes and performs oral sex, does it possibly cause genital herpes? (M, 25, HE) | |
| | **8.4. Abortion** | 36 |
| | Ex: Why do so many people have abortions? (M, 11, 2nd CBE) | |
| | Ex: If a woman gets pregnant and miscarries, is there a possibility of the next pregnancy being instantly miscarried? (F, 17, secondary school) | |
| | **8.5. Contraceptive methods** | 121 |
| | Ex: How do you put on a condom? (M, 14, 3rd CBE) | |
| | Ex: If we take the pill and stop during a month and return to taking it, is it bad for health? Because of the hormones. (F, 19, HE) | |
| | **8.6. Cancer** | 20 |
| | Ex: Can oral sex cause throat cancer? (F, 16, secondary school) | |
| | Ex: What causes prostate cancer? How do you prevent it? (M, 26, HE) | |

Bold format highlights the new subcategories created.

## 4. Discussion

Globally and given the responses to the main research question "In which areas of sexuality education do students have more interest and curiosity?", these results show many questions associated with biology, physiological, and behavioral domains. Moreover, the analysis of the global data collected revealed a large number of questions not directly related to sexuality (17.0%), as well as a considerable refusal to ask questions (10.4%). The questions not related to sexuality evidence that this exercise may have been an opportunity for children and adolescents to ask about some concerns they had, namely about familiar diseases, which they have not yet learnt about. As it was not our focus, we did not consider this group of questions in the study.

According to our content analysis, the more interesting topics for students seem to be those related to biological aspects of human sexuality, even in the questions concerning sexual behavior, because these questions were put essentially in terms of physiological reactions, including the sexual response. What children and adolescents want to know more, or where they express more doubts and curiosities, tends to involve a biological dimension. This can be interpreted based on two perspectives. On the one hand, it can be interpreted from the perspective of the traditional biomedical model of sexuality and sex, understanding sexuality as synonyms of reproductive anatomy, reproduction, and pregnancy. On the other hand, we can interpret that values, social interactions, and cultural aspects are more debatable and comfortable to discuss, while topics of intercourse, reproduction, contraception, and sexual response are more censored and threatening. These differences can also be reinforced by the curricular programs of natural sciences and biology as the unique school method to approach themes related to sexuality. A comprehensive sexuality education continues to be difficult to implement in Portuguese schools.

In the category "Violence and Staying Safe", there were no questions related to consent, privacy, or the safe use of technologies to communicate. These results reveal some issues that can be interpreted as an absence of education about risks, rights, and support, as well as a conception that it is not a sexuality issue. On the other hand, some support questions were included in the category "Skills for health and well-being", namely about the need for moments and spaces for sexuality education at school. Nevertheless, questions related to

internet use and safety appeared, and we know that this is one of the most frequent ways young people use to clarify their doubts and to communicate, often with strangers. Thus, these results indicate comprehensive sexuality education at school, covering all the eight key concepts from UNESCO [1].

Considering differences between the sexes, girls asked more questions related to sexual and reproductive health, namely reproduction and puberty, as well as on the human body and development, than boys. On the other hand, boys asked more questions about sexual behavior than girls. Additionally, boys refused to ask questions more often than girls. These results reject our first null hypothesis. The differences between boys and girls are in line with the research of Marinho and Anastácio [17], questioning adolescents about what they want to learn more in sexuality education, where boys showed more interest in topics related to sexual desire and girls showed more interest in family planning and sexual and reproductive health. This aspect underlines the idea that gender-focused programs are more effective than "gender-blind" programs in achieving health outcomes, namely the reduction in unintended pregnancies and sexually transmissible infections [1]. Despite this, sexuality education programs do not divide any activities by sex [10] and, therefore, even if the interests and concerns of boys and girls expressed in our study are different, we think that they both need to learn about reproduction, menstruation, sexual and reproductive health, and sexual desire. In this sense, we consider that boys and girls should be listened to about how they would feel more comfortable during sexuality lessons. They probably need moments to work separately by sex or gender and need other sessions including the same sexuality education programs as a way to empower and give them equal opportunities to discuss and learn about sexuality. In addition, it can reinforce the co-responsibility for safe and healthy sexuality. In this sense, future research will be needed to explore behavioral changes and health outcomes.

Our results reveal a significant difference in topics of interest in school levels, reinforcing the idea that younger children are more curious about how babies are born and develop. Young adolescents are curious about sexual behavior and abortion, middle adolescents are more interested in pregnancy prevention and sexually transmissible infections (STIs), and the late adolescents show great interest in infertility, endocrinology, and cancer of the glands of the reproductive and sexual systems (breast and prostate). In this sense, the second null hypothesis is rejected.

The categories that stand out more in our results were "The Human Body and Development", "Sexual and Reproductive Health", and "Sexuality and Sexual Behavior"; these are also those most addressed in sexuality education programs [18]. According to Holstrom [19], young people want "information about sexual pleasure, how to communicate with partners about what they want sexually and specific techniques to better pleasure their partners" (p. 282), and argued for an approach that places more emphasis on adolescent sexual and positive reproductive health [20].

The categories "Relationships", "Values, Rights, Culture, and Sexuality", "Violence and Staying Safe", and "Skills for Health and Well-being" were those in which a smaller number of questions/doubts were included. Apart from the subcategories "Violence and Staying Safe" and "Skills for Health and Well-being", the rest do not appear in the curricula of sexuality education programs [18] and this may mean that children and adolescents, even with doubts and interests in these matters, do not keep them in mind when asking questions. This proves that though all students ought to have access to a health education program containing progressive and internationally recommended sexual health education topics, too many curricula seemingly fail to deliver [18]. This happens because while there are national and international orientations, schools have a certain degree of freedom to shape the sexuality education they offer based on the perceived needs within their school community. While this is generally positive, it also opens the possibility that certain aspects of the curriculum take precedence over others or that other aspects are omitted entirely [21].

Regarding the emerging subcategories, for the category "Understanding Gender", a subcategory covering issues related to sexual orientation emerged. New subcategories related to abortion, contraceptive methods, and cancer emerged in the category of "Sexual and Repro-

ductive Health". However, in 2017, Denford [22] highlighted the promotion of condom use and other safer sex strategies, and two reviews of the literature showed that most programs focused on the physical and biological aspects of sex and on the prevention of unplanned or teenage pregnancies, the transmission of HIV or sexually transmitted infections, or violent relationships [10,23]. Some of these aspects relate to the interests of our sample, but violent relationships and safety were not mentioned as much. Although there have been advances in recent years, the emerging subcategories are in line with the ideas of other authors who defend the urgent need to do more to address issues such as abortion and female genital mutilation, which are rarely addressed [24]. Furthermore, sexual education that privileges heterosexuality [23] and marginalizes LGBTQ+-related information about healthy relationships continues to be absent from sexual and reproductive health programs [25].

Considering the developmental stage, a pattern of issues and interests was found depending on the school grade, noting that the children's interests are related to conception, birth, and the well-being of the fetus/newborn as found by Tunnicliffe and Reiss [26]. For adolescents, their interests focus on contraception, sexually transmissible infections, and relationships. The differences between sexes and school levels were significant. In this way, knowing the interests of each group, we are more able to develop sexuality education skills appropriate to the student's developmental stage, motivating them to learn better and leading to the empowerment and the reduction in risky sexual behaviors, as well as to conscious decision making to contribute to a healthy sexuality, as is suggested in the health promotion approach.

Summarizing the strengths of this study, the sexuality domains in which students have more questions were established, as well as the type of questions. The categorizations according to the UNESCO key concepts, instead of their extent, revealed a lack of some topics in sexuality education. Additionally, differences between sexes and the developmental stages and/or school grades were found. It was a past research need that now can contribute to adjusting the content of sexuality programs at school. The study also has some weaknesses, namely the limitations described below regarding the restricted region of data collection.

## 5. Conclusions, Limitations, and Future Research Works

In conclusion, based on the evidence that 91.8% of questions were situated in the three categories "The Human Body and Development", "Sexual and Reproductive Health", and "Sexuality and Sexual Behavior", most questions were related to the biological aspects of human sexuality. Additionally, this demonstrates biology/nature sciences teachers' important role in responding to students' interests and curiosities. In other words, there is a need for science education in sexuality education, without forgetting the other dimensions of a comprehensive sexuality education [1]. It is necessary to involve other teachers and interveners from diverse psychological, social, cultural, and rights backgrounds. Differences between sexes, as well as between the school grades, were verified, leading to the rejection of our two null hypotheses.

Despite national and international sexual health recommendations for a broader and more comprehensive sexuality approach [1,4], some relevant content seems to be missing in these international guidelines, to cover the emerging categories from our study to respond to the questions, doubts, and curiosities of Portuguese children and adolescents. Therefore, following the development and implementation of sexuality education programs according to the previous findings, it is vitally important to increase the likelihood that students receive the full range of sexuality education they need [22]. Furthermore, it is important to also include children and adolescents as participants in all phases of the design, implementation, and evaluation of sexuality education programs, allowing them to participate in reflections on their effectiveness as a way of ensuring a full cycle of participatory action research [27].

The results of our study can contribute to improving the content taught to children and adolescents and the quality of the activities carried out in the field of sexuality education. Health education related to sexuality is one of the best ways to prevent risk behaviors and

to promote a healthy and responsible sexuality. Moreover, these categories of questions according to sex and school grade can provide information for teachers, psychologists, nurses, and other health and education professionals who interact with children and teens. To help sexuality education professionals, we are preparing pedagogical resources based on these results.

Some limitations should be noted. Despite the inclusion of several school levels, convenience sampling was chosen. Thus, there is a possibility of selection and information bias. The sampling region of our study was located in northern Portugal, which could limit children's and adolescents' perspectives of sexuality education. To improve the generalizability of this study, we would have liked to extend this study to other regions and have a larger sample size. Furthermore, the data were collected before the COVID-19 pandemic. Although changes may have occurred in more recent years, the standards for sexuality education that were in place at that time have not recently changed. Finally, this study was not intended to evaluate individual curricula or their implementation in schools. Instead, the participants' reflections led researchers to assume that doubts and questions alone may be insufficient to translate the experiences and activities developed within sexuality education programs. Even so, the data collected, despite these limitations, provide rich material for consideration, namely for those who intend to prepare teacher training on sexuality education or to implement sexuality education programs at school, as well as to produce pedagogical resources appropriate for children and adolescents' development, clarifying the doubts of both sexes.

**Author Contributions:** Conceptualization, Z.C.A.; methodology, Z.C.A.; software, Z.C.A., R.F.A. and C.A.; validation, Z.C.A., R.F.A., C.A., M.D.G.-L. and R.B.-A.; formal analysis, Z.C.A., R.F.A. and C.A.; investigation, Z.C.A., R.F.A. and C.A.; data curation, Z.C.A., R.F.A. and C.A.; writing—original draft preparation, Z.C.A.; writing—review and editing, Z.C.A., R.F.A., C.A., M.D.G.-L. and R.B.-A.; visualization, Z.C.A., R.F.A., C.A., M.D.G.-L. and R.B.-A.; supervision, Z.C.A., R.B.-A., C.A., M.D.G.-L. and R.B.-A.; project administration, Z.C.A.; funding acquisition, Z.C.A. All authors have read and agreed to the published version of the manuscript.

**Funding:** This work was financially supported by Portuguese national funds through the FCT (Foundation for Science and Technology) within the framework of the CIEC (Research Centre on Child Studies of the University of Minho) project under the reference UIDB/00317/2020 and UIDP/00317/2020.

**Institutional Review Board Statement:** The study was conducted according to the Declaration of Helsinki and the Portuguese law for data protection, as well as approved by the Ethical Committee of the University of Minho (CEICSH/007/2020).

**Informed Consent Statement:** Informed consent was obtained from all subjects involved in the study.

**Data Availability Statement:** The data presented in this study are available on justified request to the corresponding author.

**Acknowledgments:** The authors acknowledge Vânia Beliz, a doctoral student under the supervision of the contact author, Zélia Anastácio, for document preparation, for ethical procedures, and support in data collection.

**Conflicts of Interest:** The authors declare no conflict of interest.

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
