# Peer review of "Interests and Curiosities about Sexuality of Children and Adolescents from Northern Portugal"

_sexes, doi:10.3390/sexes4020020_

Round 1

Reviewer 1 Report

I found the manuscript quite interesting and the authors can find my concerns as follow

The introduction needs to be improved or rewritten. I read that the authors integrally reported a sentence from “European Expert Group on Sexuality Education. Sexuality education”. I do not agree and the authors need to introduce the topic in an original way, avoiding passively reporting statements written by other authors.

The aims and Hypotheses need to be stated in a more specific way. Indeed the authors stated, “The aim of this research was to make a survey of students’ interest along the school”- according to this aim, the principal purpose of the study is the development of a survey or a test that allows to study the interest of children in sexuality. However, this type of procedure is very complex, and several guidelines have been published. Moreover, I suggest adding specific info from previous studies about sexual education in children and adolescents. It is crucial to introduce the aims.

This was a transversal study, having the data been collected near each subject in a single moment. (Line 76)- This sentence is not clear. Indeed, I suppose that the authors aim at comparing different populations in terms of the age. This should be explained in the last part of the introduction.

Lines 91-93. The formal ethical approval (code n.) is needed. Since the study involved children the procedure about informed consent should be added (maybe given by parents or legal guardian of a minor). Moreover, the privacy is also important. Sex research usually follows a strict privacy protocol , please explain and add more info about.

Line 129-“In order to increase the validity of the  categorization it was done by three researchers separately and after compared together. No great differences were observed.” The authors explained that the methodology was qualitative, but they added that “No great differences were observed.” The authors need to be more exhaustive about this point. According to me it is possible to quantify the between researchers agreement.

I suggest assessing sex differences x school degree. Moreover, Chi sqr. Correction and effect size need to be reported.

“ Searching for any association between the categories of questions and the school 179 grade, the Chi square test indicated a significant relationship (2 (36) = 908.675, p < .001).” This analysis and result is not clear.

The results are very interesting. Indeed, children showed interest for the normative biological information about sex, but adolescents displayed the need to receive more info about endocrinology and cancer. I performed a rapid epidemiological search  (Pubmed)  about the health situation in Portugal. Cardiovascular diseases and Cancer are the principal causes of death in Portugal between 1980-2010. I suppose that adolescents were more familiar with issues related to breast carcinomas or testicular cancers. It is only a hypothesis, hoping to be helpful.

Reviewer 2 Report

The paper entitled “Children and Adolescents’ Interests in Sexuality Issues”, aims to examine Portuguese Primary School, Secondary School and Higher Education students’ interest and curiosity in relation to sexuality. At the methodological level, 741 students were surveyed in 32 classes ranging from Primary to higher education in the northern region of Portugal. The students were asked to write anonymous questions about their interest in sexuality since students could feel uncomfortable putting questions orally. The authors categorized the information obtained from students' answers according to UNESCO's eight key concepts for sex education. Although, the authors created two additional categories. Moreover, this categorization was done separately by three researchers that converged on the same conclusions.

The authors concluded that students’ interest was principally related to biological aspects of human sexuality. However, girls were more interested in sexual and reproductive health, while boys were more curious about sexual behaviour. Boys showed more interest in topics related to sexual desire and girls showed more interest in topics related to family planning.

I have three concerns about this paper.

First, according to the authors, the methodology was predominantly qualitative because they collected data through open questions, using a format A5 white page. However, I disagree with the authors. In my opinion, the authors used a quantitative methodology. Moreover, the analysis of the data is quantitative and not qualitative. The authors used a format A5 white sheet of paper where students had to write their age, school grade and gender. After that, students were asked to write anonymous questions concerning human sexuality. Therefore, the authors did not use a qualitative methodology, but a quantitative methodology. The A5 white sheet was a kind of questionnaire that included three closed-ended questions and an open-ended question concerning students’ interest in sexuality.

Second, according to the authors, the sample covered 741 students (47,7% females and 52,3% males). However, 169 students did not write any questions. Therefore, 169 questionnaires did not have any kind of useful information. Accordingly, the sample covered a total of 572 students, not a total of 741 students.

Finally, the authors concluded that girls’ interests and curiosity in relation to sexuality were different from boys’ interests and curiosity. However, on page 10 the authors conclude that “, even if the interests and concerns of boys and girls expressed in our study are different … we consider boys and girls should include the same sexuality education programs”. The authors should argue about this issue more in-depth.

This paper needs moderate editing of English language. 

Reviewer 3 Report

This is significant research containing data that has potential use as reference for many other research works. The research design with its methods and limits is clearly spelled out. The results are adequately discussed.

It might be better to briefly explain the choice of "sexuality education" over "sex education" or "sexual education"

Were the questionnaires in Portuguese or in English?

Are the questions in Tables 4-10 originally in English or are they translations? If the latter, a grammatical check is suggested.

The manuscript needs thorough grammatical check.

Some suggested corrections (only samples):

Line 161. "boys were the ones who"

Line 243. "were the two other subcategories that emerged" (?)

Lines 247-249 need improvement.

Line 252. "are evidence"

Line 254 "which they had not yet had opportunity to discuss"

Line 257. "those related to"

Line 299. "babies are born"

Line 353. "we can conclude that most questions"

Lines 354-358 need to be improved.

Reviewer 4 Report

Children and Adolescents’ Interests in Sexuality Issues

This is a very interesting and relevant contribution to Sexes. I believe the article is well written and scientifically sound. Results are compelling with relevant implications to sex education in Portugal. Still, I believe some minor contribution would improve the overall quality of the article.

1.     Please provide more results from previous research in the introduction section.

2.     Please include an implications sections, focusing on education policies in schools, teacher’s training and other agents of the educational community (family, psychologists, school staff, etc.)

Best wishes.

Reviewer 5 Report

This study derived major directions and issues for sexuality education based on a gender survey of northern Portuguese teenagers. Teenagers were asked to write random questions on A5 paper to obtain somewhat superficial survey results, and it would have been more realistic to write down concerns rather than questions. Anyway, the results of this study are believed to have provided very important basic data for future sexuality education for Portuguese teenagers.

In order to improve the completeness of this paper, I hope to revise the following.
-As the authors of this study acknowledged, the subject of this study is limited to teenagers in northern Portugal, so it is somewhat difficult to generalize the research results. Therefore, it is better to specify the name of the country "Portugal" in the title of the paper.
-Please specify the survey period in Section 2.2.
- At the end of Chapter 4, it is necessary to present the results summarized by grade and gender of teenagers in tabular form.
-Since the limitations of this study in Chapter 6 are a kind of weakness of this study, it is better to revise the title of Chapter 5 to "Conclusion and Future Research Works" and present the contents of Chapter 6 as future research works.

Round 2

Reviewer 1 Report

Unfortunately, and as I wrote in my previous appraisal, sentences like “We only found some work in  books from Ariza et al. [8] and Barragán [9] from Spain” are more similar to a response to a reviewer, not a scientific motivation.

Similarly, in the discussion “We also conclude by the diferences ”, “We believe..” I advise the authors to highlight the weaknesses and strengths of the present study. Please, despite I am not qualified to assess the English, but, please, the English language needs to be improved.

“Before the project activities, teachers and school principals gave consent  and informed the parents or tutor of children and adolescents less than 18 years old.” I suppose that with participant < 18 years old in Europe, the legal tutor or the parents need to read and sign the informed consent. http://fra.europa.eu/en/publication/2019/child-participation-research. Here, you can find the regulation for each country of UE, including Portugal. According to me, there is a misleading due to the English language, because the study received a formal approval by Ethical committee, but the authors need to clarify this point.  

Author Response

Dear Reviewer,

We want to thank you for your contribution to improving our paper. We review all the text, complete some ideas and justify some options. So, we hope to be in accordance with your recommendations. 

See attach file for details.

Best regards
